# Predictive Modeling of Long-Term Care Needs in Traumatic Brain Injury Patients Using Machine Learning

**DOI:** 10.3390/diagnostics15010020

**Published:** 2024-12-25

**Authors:** Tee-Tau Eric Nyam, Kuan-Chi Tu, Nai-Ching Chen, Che-Chuan Wang, Chung-Feng Liu, Ching-Lung Kuo, Jen-Chieh Liao

**Affiliations:** 1Department of Neurosurgery, Chi Mei Medical Center, Tainan 711, Taiwan; ronaldowen@gmail.com (T.-T.E.N.); gary12223@hotmail.com (K.-C.T.); wangchechuan@gmail.com (C.-C.W.); 2Center of General Education, Chia Nan University of Phamacy and Science, Tainan 717, Taiwan; 3Department of Nursing, Chi Mei Medical Center, Tainan 711, Taiwan; dreampatty11@gmail.com; 4Department of Medical Research, Chi Mei Medical Center, Tainan 711, Taiwan; chungfengliu@gmail.com; 5College of Medicine, National Sun-Yat-Sen University, Kaohsiung 805, Taiwan; 6Department of Neurosurgery, ChiaLi Chi Mei Medical Hospital, Tainan 722, Taiwan

**Keywords:** traumatic brain injury, long-term care, machine learning models, predictive analysis, Random Forest, SHAP analysis

## Abstract

Background: Traumatic brain injury (TBI) research often focuses on mortality rates or functional recovery, yet the critical need for long-term care among patients dependent on institutional or Respiratory Care Ward (RCW) support remains underexplored. This study aims to address this gap by employing machine learning techniques to develop and validate predictive models that analyze the prognosis of this patient population. Method: Retrospective data from electronic medical records at Chi Mei Medical Center, encompassing 2020 TBI patients admitted to the ICU between January 2016 and December 2021, were collected. A total of 44 features were included, utilizing four machine learning models and various feature combinations based on clinical significance and Spearman correlation coefficients. Predictive performance was evaluated using the area under the curve (AUC) of the receiver operating characteristic (ROC) curve and validated with the DeLong test and SHAP (SHapley Additive exPlanations) analysis. Result: Notably, 236 patients (11.68%) were transferred to long-term care centers. XGBoost with 27 features achieved the highest AUC (0.823), followed by Random Forest with 11 features (0.817), and LightGBM with 44 features (0.813). The DeLong test revealed no significant differences among the best predictive models under various feature combinations. SHAP analysis illustrated a similar distribution of feature importance for the top 11 features in XGBoost, with 27 features, and Random Forest with 11 features. Conclusions: Random Forest, with an 11-feature combination, provided clinically meaningful predictive capability, offering early insights into long-term care trends for TBI patients. This model supports proactive planning for institutional or RCW resources, addressing a critical yet often overlooked aspect of TBI care.

## 1. Introduction

Traumatic Brain Injury (TBI) stands as a significant global health concern, with an estimated annual incidence ranging from 27 to 69 million worldwide [1]. According to the Centers for Disease Control and Prevention (CDC), in the United States, TBI cases witnessed an increase from 521 to 824 per 100,000 individuals between 2001 and 2010 (CDC, 2011). Similarly, in Taiwan, the incidence was reported at 220.6 per 100,000 person-years from 2007 to 2008 [2]. TBI-related mortality rates are approximately 17.7 deaths per 100,000 in the U.S. (53,000 deaths) [3] and range from 26.15% to 36.36% in Taiwan for patients undergoing brain surgery between 1997 and 2007 [4,5].

TBI results in enduring cognitive, physical, and behavioral impairments, necessitating prolonged healthcare and disability assistance services [6]. The estimated healthcare cost for non-fatal TBI among MarketScan enrollees in the USA in 2016 amounted to USD 40.6 billion [7]. Consequently, TBI not only significantly impacts patient well-being but also imposes a substantial burden on social, economic, and healthcare resources (CDC, 2016). Hence, accurate predictions and contributions to a more nuanced understanding of the factors influencing TBI outcomes are crucial.

Several studies have identified a range of predictive factors that may significantly influence the outcomes of individuals with TBI in both acute and chronic stages. These factors include age [8,9], sex [9,10], obesity [11], Glasgow Coma Scale (GCS), and pupil reactivity [12,13], computed tomography (CT) findings [14,15], surgery [8,16], injury severity [17,18,19,20], vasopressor use [21], endotracheal tube intubation [22], intracranial pressure monitoring use [23], and hypnotic-sedative drug use [24]. Therefore, establishing how to effectively utilize these feature variables for predicting outcomes is a crucial issue.

The objective of returning home (RH) holds significant importance for survivors of the intensive care unit (ICU), as emphasized by Li Y [25]. Research indicates that approximately 20% to 38% of ICU survivors following TBI fail to achieve discharge to their homes [26]. Various studies have explored predictive factors for RH in TBI patients, including age, heart rate, platelet count, D-dimer, and GCS score, as reported by Yabuno S [27]. Additionally, Leitgeb identified age and GCS score as significant predictors [28].

Additional factors influencing the likelihood of a positive outcome with RH in TBI patients include the length of hospital stay [29], the Functional Status Score for the ICU [30], the Acute Physiology and Chronic Health Evaluation II score (APACHE II) [31], and the Coma Recovery Scale-Revised (CRS-R) score at discharge [32]. However, the complexity of TBI outcomes and the limitations of traditional statistical methods may be insufficient in capturing the interactions among various clinical variables, especially with multiple interacting variables and non-linear relationships [33]. Therefore, the quest for a method that offers the potential to uncover intricate patterns, provide accurate predictions, and contribute to a more nuanced understanding of the factors influencing TBI outcomes is crucial.

Machine learning, a subset of artificial intelligence (AI), proves invaluable in clinical prediction and the discovery of novel prognostic markers, due to its capability to detect interactions among numerous attributes [34]. However, a notable impediment to its widespread clinical application is the lack of explanation. Explanatory AI (XAI), such as SHAP (SHapley Additive exPlanations), becomes crucial for enhancing the interpretability of machine learning models and understanding essential clinical features in predicting diseases or patient outcomes [35]. The importance of features is a critical aspect of comprehending the relevance of clinical features in the prediction process, and various XAI techniques contribute to elucidating this importance.

While recent studies have applied machine learning to predict outcomes in neurological disorders such as aneurysmal subarachnoid hemorrhage [36] and ischemic stroke [37], and machine learning has been applied to TBI outcomes, the primary focus has been on mortality or morbidity [38,39,40,41]. Its application in predicting the likelihood of transferring to a long-term care center after TBI is not well-established. Therefore, developing new AI prognostic prediction models specifically for this outcome is worthwhile.

In this study, we aim to address the pressing challenges in predicting outcomes for ICU patients with TBI who are unlikely to return home, but transfer to long-term care center. Our specific objectives are to develop and validate machine learning-based prognostic models, including Logistic Regression (LR) [42], Random Forest (RF) [43], LightGBM [44], and XGBoost [45], and Multilayer Perceptron (MLP) neural networks [46], which predict the risk of transferring TBI patients to a long-term care center, using easily obtainable clinical data. To enhance the interpretability of our models, we employ the SHAP [47,48] technique to identify and explain the key clinical features influencing these predictions. By focusing on this underexplored aspect of TBI outcomes, we hope to provide valuable insights that can inform clinical decision-making and contribute to the development of targeted interventions for this vulnerable patient population.

## 2. Materials and Methods

### 2.1. Ethics Consideration

Approval for this study was granted by the Institutional Review Board (IRB) of Chi Mei Medical Center under the reference number 10911-006. All procedures were conducted by the authors in strict adherence to applicable laws and regulations, performed in accordance with relevant guidelines. Given the retrospective nature of the study, the ethics committee decided to waive the requirement for obtaining informed consent.

### 2.2. Study Flow Chart and the Content of the Prediction Device

Our study followed the guidelines specified in the Transparent Reporting of a Multivariable Prediction Model for Individual Prognosis or Diagnosis (TRIPOD) standard. Figure 1 illustrates the flowchart detailing the ML training process and its integration into the hospital system for TBI patients in ICU. Various models, including LR, RF, LightGBM, XGBoost, and Multilayer Perceptron (MLP) neural networks were trained on 70% of the data and validated on a 30% test set through random splitting. To mitigate concerns of overfitting that might arise from a small dataset, we employed the cross-validation technique to build the models.

In this study, we excluded deceased patients, as the primary focus was on predicting long-term care needs for TBI survivors. For the remaining dataset, we opted not to impute missing values, due to their significant clinical implications in a healthcare context. Imputation of variables such as SOFA scores, FiO2, or Glasgow Coma Scale (GCS) may compromise the clinical integrity of the data, as these features are critical indicators in patient management and decision-making.

Therefore, we chose to exclude records with incomplete clinical data, as reflected in the updated Figure 1 Flowchart. The extent of missingness for each variable is summarized in the RTable (Appendix A), where variables such as SOFA (24.55% missing) and FiO2 (10.94% missing) were noted. This approach ensured that only high-quality, complete clinical data were used for model development, thus enhancing the reliability and robustness of the machine learning predictions.

To address the imbalance in the dataset, characterized by more negative cases (survival) than positive cases (mortality), we applied the Synthetic Minority Oversampling Technique (SMOTE) [49] to achieve equal representation during the final model training with each algorithm. To evaluate the stability and scalability of the proposed models, we conducted both cross-validation and scalability testing. Stability was assessed using 5-fold cross-validation, where the mean and standard deviation of Accuracy [50], Sensitivity, Specificity [51], and Area under the Curve (AUC) [52] of the Receiver Operating Characteristic curve (ROC) were analyzed across different training and validation splits. Figure 2 illustrates our AI prediction device for TBI in the ICU, providing insight into the system’s architecture and modules.

### 2.3. Schematic Representation and Configuration of the Prediction Apparatus

Our investigation adhered to the guidelines outlined in the Transparent Reporting of a Multivariable Prediction Model for Individual Prognosis or Diagnosis (TRIPOD) standard. Figure 1 delineates the diagram portraying the machine learning (ML) training procedure and its seamless integration into the hospital system for patients with traumatic brain injury (TBI) in the intensive care unit (ICU). Diverse models, encompassing Linear Regression (LR), Random Forest (RF), LightGBM, XGBoost, and Multilayer Perceptron (MLP) neural networks, underwent training on 70% of the dataset and validation on a separate 30% test set following random partitioning. To address potential overfitting concerns stemming from a limited dataset, we employed the cross-validation technique in constructing the models.

In order to counterbalance the dataset’s imbalance, marked by a surplus of negative cases (no transfer to long-term care center) compared to positive cases (transfer to long-term care center), we implemented the SOMET. This technique was instrumental in achieving equitable representation during the final model training across all algorithms. Figure 2 offers an illustration of our artificial intelligence (AI) prediction apparatus designed for predicting TBI in the ICU, shedding light on the architecture and components of the system.

### 2.4. Defining Enrollment Period, Inclusion and Exclusion Criteria in the Current Study

This study retrospectively enrolled 3958 patients aged 20 and above with Traumatic Brain Injury (TBI). These individuals were admitted to the Intensive Care Unit (ICU) at Chi Mei Medical Center in Tainan, Taiwan, between 1 January 2016 and 31 December 2021. The electronic medical records were screened using the following diagnostic codes: ICD-9: 800*–804*, 850*–854*, 959.0, 959.01, 959.8–959.9, ICD-10: S00*–T07*. Primary outcome is defined as “admitted to long-term care center excluding death”. Data with missing values, extreme outliers, and those who had deceased were also excluded (1938 patients).

### 2.5. Feature Selection for Predicting Traumatic Brain Injury Transfer to Long-Term Care in ICU

Initially, a collection of 44 feature variables, routinely acquired and measured, was gathered upon admission to the intensive care unit. These variables were deemed crucial for predicting the progression of Traumatic Brain Injury (TBI). The initial 44 features were selected by experts based on their knowledge of the subject matter, previous research findings, and clinical relevance. All of the features have been analyzed and identified as encompassing a range of predictive factors that may significantly impact the outcomes of individuals with TBI across both the acute and chronic stages. Subsequent feature reduction or recombination was performed based on clinical significance (*p* < 0.05) and Spearman correlation coefficient (absolute value > 0.1 or >0.2) analysis to select the necessary parameters. These criteria were applied to construct a fitting model for predicting the likelihood of transfer to the long-term care center.

### 2.6. Assessing Machine Learning Models for Predicting Traumatic Brain Injury Outcomes in ICU

In our study, we deliberately selected commonly used and stable machine learning algorithms, including LR, RF, LightGBM, XGBoost, and MLP. These algorithms are widely acknowledged in the medical field for their robustness, interpretability, and strong predictive capabilities, which align with the practical goals of our study. To evaluate the performance of our machine learning models, we implemented and compared multiple commonly used algorithms. Additionally, we incorporated a stacking ensemble method, which combines predictions from these base models to potentially improve overall performance. To evaluate the quality of these models, we utilized metrics such as Accuracy [50], Sensitivity, Specificity [51], and AUC [52]. Ultimately, we selected the model with the highest AUC for clinical implementation, as this metric reflects the overall discriminative ability of the model and ensures the best balance between sensitivity and specificity. A higher AUC value indicates a superior model, demonstrating its ability to distinguish between the two classes across different threshold levels.

The DeLong test specifically compares areas under correlated ROC curves, indicating significant differences in performance between the models [53].

To gain insights into how each feature contributes to the associated outcome, we employed SHAP (SHapley Additive exPlanations) analysis, a widely used technique for explaining the importance of clinical features in predicting various diseases or patient prognosis.

### 2.7. Statistical Analysis

The statistical analysis for this study was conducted using the SPSS software (Version 15, SPSS Inc., Chicago, IL, USA). To assess the significance of various variables, we employed specific statistical tests based on the nature of the data and research questions. For continuous variables, such as physiological measures and scores, we utilized the independent *t*-test to compare means between two groups. The Chi-square test was employed for the analysis of categorical variables. Additionally, Spearman’s correlation method was used to evaluate the strength and direction of relationships between each feature and mortality. All significance testing was performed with a predetermined alpha level of 0.05, and *p*-values below this threshold were considered statistically significant.

## 3. Results

### 3.1. Demographics and Clinical Pictures in Patients with or Without Transfer to Long-Term Care Center After TBI

A total of 2020 patients were retrospectively included, from the electronic medical records system of Chi-Mei Hospital, consisting of 1285 (63.61%) males and 735 (36.39%) females, with an average age of 63.26 years (mean ± SD: 17.76). Notably, 236 patients (11.68%) were transferred to long-term care centers. In comparison with the non-transferred group to institutions or respiratory care wards, Table 1 suggests that individuals transferred after traumatic brain injury (TBI) exhibited more severe medical conditions, evident in various physiological measures, medical interventions, and medical history. Among the 44 features, 27 (in italics) showed significant differences between the groups (*p*-value < 0.05). Due to an imbalanced outcome class in the dataset, the current study employed the Synthetic Minority Over-sampling Technique (SMOTE) for model training.

### 3.2. The Correlation Between Features and Transfer to Long-Term Care Center After TBI (Spearman Correlation Coefficient)

To quickly select the proper parameters for machine learning, we conducted a correlation analysis using the Spearman correlation coefficient [49,52]. Table 2 shows that 11 features had an absolute value of coefficients greater than 0.2 (bold), and 7 had values greater than 0.1 but less than 0.2 (italic). All the correlations in our dataset are less than 0.3. The top seven features were Muscle_RLE, Muscle_LLE, GCS_M, GCS_V, APACHE II, SOFA, and GCS.

### 3.3. Combinations of Features According to Clinical Significance and Spearman Correlation Coefficient

Four sets of feature combinations were generated, based on clinical significance, Spearman correlation coefficient values, and expert judgment with neurologists or other relevant medical experts who could provide insights into which variables are crucial in the context of TBI. These sets comprised 44 features (encompassing all features in the study), 27 features (identified as clinically significant with *p* < 0.05 between two comparison groups), 18 features (clinically significant with *p* < 0.05 between two comparison groups, along with Spearman correlation coefficient greater than 0.1 or less than −0.1), and 11 features (clinically significant with *p* < 0.05 between two comparison groups, coupled with Spearman correlation coefficient greater than 0.2 or less than −0.2) for training the machine learning model. Detailed information on these sets is presented in Table 3.

### 3.4. Area Under the Curve (AUC), for Transfer to Long-Term Care Center Prediction

Using ROC analysis and AUC [50,51] calculations, we identified optimal predictive models for the risk of transfer to an institute or respiratory care center based on different feature combinations. Across four machine learning models, the best-performing model consistently achieved an AUC exceeding 0.8 (Figure 3a–d).

The LightGBM model excelled with an AUC of 0.813 among the 44 features. In the 27-feature combination, XGBoost achieved the highest AUC at 0.823. For the 28-feature combination, Random Forest outperformed others with an AUC of 0.819. In the 11-feature combination, Random Forest again had the highest AUC of 0.817. Details are in Table 4A–D. Appendix A shows the hyper-parameter range for experiments, detailing the ranges of the hyperparameters explored.

Appendix A showed that the models consistently maintain robust performance, with minimal variation, confirming their stability. For scalability analysis, we tested the models with varying sample sizes and observed their performance through learning curves. The learning curve for the 11-feature Random Forest model (Appendix A) indicate that the models converge effectively as the training data size increases, showing no significant overfitting or underfitting trends, as in Appendix A.

The stacking method integrates outputs from individual models into a meta-model to generate final predictions. We assessed all models using metrics such as Accuracy, Sensitivity, Specificity, and AUC, as summarized in Table 4. While stacking demonstrated competitive performance and achieved the highest AUC in several scenarios, the improvement over the best-performing single model was minimal. Moreover, stacking required significantly more computational resources and training time, due to the combination of multiple base models. This trade-off between performance and resource demand was an important factor in our final model selection. Ultimately, we selected the single algorithm with the highest AUC (excluding stacking) for implementation, ensuring that the chosen model strikes a balance between accuracy, interpretability, and practical applicability in clinical settings.

### 3.5. The DeLong and Other Tests for the Best ML Models in Different Feature Combinations

To select the optimal model for clinical practice, we conducted a comparative analysis of the AUC values for the best machine learning models across different feature combinations. As presented in Table 5, XGBoost, utilizing 27 features, achieved the highest AUC (0.823), outperforming Random Forest with 11 features (AUC: 0.817). These results suggest that XGBoost, incorporating 27 features, demonstrates slightly superior overall discrimination.

To statistically assess the performance differences among the models, we employed the DeLong test [52]. The *p*-values derived from the DeLong test were examined to determine whether there were significant variations in AUC across the models. However, none of the comparisons yielded statistically significant results, implying that the models did not exhibit significant differences in AUC.

It is noteworthy that XGBoost with 27 features not only exhibited the highest AUC, but also demonstrated the highest sensitivity (0.761), indicating commendable performance in accurately identifying positive cases. On the other hand, Random Forest with 11 features achieved the highest specificity (0.753), showcasing its proficiency in accurately identifying negative cases.

### 3.6. Feature Importance of AI Algorithms Using SHAP Analysis

Next, we applied SHAP analysis to evaluate the impact of each feature on the model’s output, providing insights into the importance of individual features for outcome predictions. In Figure 4a and Figure 5a, the color of the SHAP plot represents the size of the original feature values, with red indicating positive variable values and blue indicating negative ones.

In a Random Forest model with 11 features, the significance order of the features is as follows: Muscle_RLE, Endotracheal intubation, APACHE II score, GCS_V, SOFA score, Muscle_LUE, Muscle_LLE, Muscle_RUE, FiO2, GCS_M, and GCS. This ranking implies that features listed earlier have a greater impact on the model’s predictions. For instance, “Muscle_RLE” holds the highest importance, while “GCS” is considered relatively less important (see Figure 5a,b).

In an XGBoost model with 27 features, the top eleven features are as follows: Muscle_LLE, Muscle_RLE, APACHE II score, Muscle_LUE, Endotracheal intubation, GCS_V, SOFA, Age, GCS, Muscle_RUE, and FiO2. Except for age and GCS-M, the other features are the same as in the Random Forest model.

### 3.7. Presentation of AI Interface in Real-World Clinical Applications at Chi Mei Hospital Healthcare System

Following our analyses, we have determined that the Random Forest model, using 11 features, is simpler and lightweight. We have integrated it into the hospital system to aid clinical staff and enhance communication with patients’ families.

The “Original” column displays data from ICU admission, while the “Adjust” column allows observers to modify feature values, offering insights into their impact on the risk of transfer to an institute or respiratory care center (see Figure 6).

To assess the practical feasibility of the proposed models for clinical applications, we conducted a runtime analysis that evaluated both training and inference times. The results showed that the training times for the models ranged from 30 to 90 min, reflecting the computational complexity of each algorithm. Despite variations in training duration, all models achieved rapid inference speeds, with prediction times under 2–3 s per case.

## 4. Discussion

### 4.1. Novelty of Current Study

This is the first study to combine feature variables to predict the risk of transferring to a long-term care center using an AI model after TBI. We adopted a machine learning approach, which is less constrained by the assumptions of linear models (such as normality and independence). We found that the tree-based RF machine learning model showed clinically meaningful predictive capability with 11 feature combinations. Moreover, this approach has been implemented in a clinical system, and aids in clinical decision-making, planning by the medical team, and shared decision-making with patients. This study employed common and stable ML models. The primary objective was derived from clinical needs, with the practical aim of developing the model into predictive software integrated with the existing HIS to provide clinical decision support. Thus, it has significant contributions. We believe the development of these prognostic models has the potential to revolutionize patient management, enhance resource allocation, and improve healthcare outcomes.

### 4.2. Demographics and Clinical Pictures

Our demographic analysis reveals that 11.68% (236/2020) of TBI patients did not return home upon discharge, a notably lower percentage than reported in previous studies (20% to 38% ICU survivors) [26]. This difference may stem from variations in patient severity; our study focused on a broader GCS range (mean 12.27 ± 4.11) compared to previous studies concentrating on moderate to severe cases (GCS 3–12).

Consistent with existing research, our findings highlight that factors such as higher heart rates [27], older age and lower GCS [27,28], and elevated APACHE II scores [31] are associated with a reduced likelihood of returning home. Additional contributors include abnormalities in pupil light reflex, muscle power, SOFA score, drug usage, and specific comorbidities like cerebrovascular issues and pneumonia. A comprehensive assessment of 27 features (see Table 1) indicates significant statistical differences between transferred and non-transferred groups.

These insights underscore the need to consider TBI severity and a comprehensive set of clinical indicators when predicting discharge outcomes. The inclusion of diverse factors in our study contributes to a nuanced understanding of the determinants influencing post-TBI discharge destinations

### 4.3. Correlation Analysis

To enhance predictive modeling based on available features, we assessed correlations using the Spearman coefficient. All correlations in our dataset are weak (|r| < 0.3), indicating subtle relationships between variables. While these correlations are not strong, they offer insights into variable relationships.

In line with prior research, the top 18 features are all predictors of outcome after TBI (12, 13, 16, 21, 22, 23, 24, 27, 28, 31). Age exhibits a significant (*p* < 0.001) difference between groups in Table 1, but the weak correlation (r = 0.08) in Table 2 suggests a minor influence on the No Return Home group. This implies age may modestly impact a patient’s ability to return home, possibly moderated by unexplored variables.

Positive correlations observed in the Spearman analysis with features such as FiO2 and endotracheal intubation underscore the critical role of respiratory care, emphasizing the significance of early monitoring of oxygen levels. Additionally, correlations with severity scores (APACHE II, Sequential Organ Failure Assessment (SOFA score)) suggest the necessity of integrating these scores into intervention planning. Furthermore, the positive correlations with vasopressors, ICP, and sedative–hypnotic drugs highlight the importance of vigilant monitoring during interventions.

On the other hand, negative correlations with GCS components underscore the crucial role of neurological assessments, especially for patients with lower GCS scores requiring intensive care. Negative correlations related to muscle strength (Muscle_RLE, Muscle_LLE, Muscle_LUE, and Muscle_RUE) emphasize the predictive role of muscle strength in transfers. This emphasizes the need for early, tailored rehabilitation programs to enhance recovery.

In summary, our findings indicate that a multidisciplinary approach, involving neurologists, intensivists, respiratory therapists, and rehabilitation specialists, ensures comprehensive care tailored to the specific needs of TBI patients.

### 4.4. Feature Combinations

Most previous studies on AI have utilized a single set of features to assess the performance of AI models [40,41]. Building upon our earlier research [54], we opted for multiple combinations of features, comprising a total of four sets of data. The generation of these feature sets was guided by considerations of clinical significance, Spearman correlation, and expert judgment, reflecting a meticulous approach to feature selection.

The inclusion of expert input ensures that the chosen variables are in line with clinical understanding, thereby enhancing the model’s interpretability and applicability in a healthcare context. This careful feature selection process not only improves the interpretability of the model, but also instills confidence in clinicians that the selected variables align with clinical understanding, providing actionable insights for decision-making in a real-world setting.

### 4.5. Comparisons on Overfitting and Generalization Issues of the Models

To evaluate and address potential overfitting and generalization issues, we employed 5-fold cross-validation to assess model performance across different data partitions. As shown in Appendix A, the results include the mean and standard deviation for Accuracy, Sensitivity, Specificity, and AUC, which provide insights into model stability. Notably, Random Forest, LightGBM, and XGBoost demonstrated low standard deviations across folds, indicating strong robustness and consistent performance. For instance, Random Forest, with 11 features, achieved an AUC of 0.956 ± 0.038, highlighting its ability to generalize across diverse datasets. Similarly, the Stacking ensemble method exhibited stable performance with minimal variance, suggesting its potential to mitigate overfitting by leveraging complementary strengths of individual models. However, we acknowledge that this study relies solely on electronic medical records from a single medical center in Taiwan, which may limit the generalizability of our findings to other populations or healthcare settings. While our cross-validation results support the robustness of the models within the current dataset, future studies should include external validation with data from multiple centers, to further confirm generalizability.

### 4.6. AUC Analysis and Delong Test

The ROC analysis and AUC calculations showcased the discriminative capabilities of various machine learning models—LR, RF, LightGBM, XGBoost, and Multilayer Perceptron (MLP)—across different feature combinations. The consistently moderate AUC values (>0.8) [Bowers, A.J., 2019] in our study, affirm the efficacy of all tested AI models in predicting transfers (refer to Table 4 and Table 5). The differences in feature sets generated by different ML methods could be due to algorithmic differences and data preprocessing methods.

The Delong test [52] was conducted to assess the statistical significance of the comparisons, and the results indicated that none of the comparisons yielded statistically significant differences in AUC (see Table 5). This suggests that the models did not exhibit notable variations in their discriminative performance. Consequently, these findings imply that all models effectively discriminate between patients who will be transferred and those who will not.

### 4.7. SHAP Analysis Evaluating Feature Importance

Using SHAP analysis (Figure 5 and Figure 6), we identified that all the features in the Random Forest model (11 features) and XGBoost model (27 features) are clinically significant for the outcome. Except for age and GCS-M, the remaining features are consistent among the top eleven features in both the RF and XGBoost models. SHAP analysis exposes the impact of individual features on model predictions, and understanding the significance of the order of these features can assist clinicians in prioritizing interventions [53].

For example, the top five features in terms of importance in the RF model are Muscle_RLE, Endotracheal intubation, APACHE II score, GCS_V, and SOFA score. When a patient has poor right upper-limb function and impaired speech, these conditions can lead to greater dependency on assistance for daily activities. Additionally, if the patient has undergone endotracheal intubation and has higher APACHE II and SOFA scores, it indicates greater severity of illness and more affected organs. This will require more intensive care and support, impacting the patient’s ability to be discharged home successfully without needing transfer to a long-term care facility.

### 4.8. Considering Model Selection: Which Is Preferable, Random Forest or XGBoost?

In addition to the five algorithms, we incorporated a Stacking method to compare the performance improvement of an ensemble of all algorithms (as shown in Table 4). While the stacking model slightly outperformed the best single model in terms of AUC, the improvement was marginal. For instance, in the 11-feature combination, the stacking model achieved an AUC of 0.820 compared to 0.817 for the Random Forest model. However, stacking requires substantially more computational resources and training time, due to the integration of multiple base models into a meta-model.

Considering these factors, we prioritized the single model with the highest AUC (excluding stacking) for clinical use. This decision ensures that the model is both efficient and practical for real-world implementation, particularly in resource-constrained clinical environments where quick predictions and model interpretability are essential

In the present study, XGBoost (AUC = 0.823) demonstrates a slight advantage in these metrics compared to Random Forest (AUC = 0.817), though the superiority may not be highly significant. Therefore, in the current study, we chose Random Forest (11 features) as the final predictive model, based on its well-rounded performance in terms of accuracy (0.752) and a balance between sensitivity (0.746) and specificity (0.753). Additionally, the AUC value of 0.817 suggests that the model has good discriminative ability. The involvement of the fewer 11-feature model is another important point of consideration, based on the fact that cost-effectiveness and convenience would be more practical for implementation.

### 4.9. Real-World Application

The incorporation of the Random Forest model into the hospital system represents a crucial step toward real-world clinical use. Its simplicity and efficiency make it practical for daily application. This software enhances the ability to adjust and interact with prediction functions, allowing for manual adjustment of parameter values to re-predict outcomes. It can simulate changes in a patient’s physical condition to see if it affects prognosis. For example, it can show how a decrease in FiO2 increases risk. This enhances the contribution of our research.

We conducted a runtime analysis to evaluate the computational efficiency of the pro-posed models, focusing on both training and inference times. The results indicate that the training time for individual algorithms ranged from approximately 30 to 90 min, depending on the complexity of the model and the size of the dataset. For inference, all models demonstrated excellent efficiency, with prediction times consistently under 2–3 s per case.

The AI model helps close gaps in TBI care by making decisions more personalized and data-driven. It predicts long-term care needs, helping healthcare teams plan re-sources like beds and staff, to avoid delays and ensure timely care. Linked to the hospital’s EHR system, it gives real-time predictions to support discharge planning, coordinate with care centers, and prevent issues from long hospital stays. It also helps patients and families by providing tailored insights to set realistic expectations and plan ahead. This proactive approach makes care smoother, reduces rehospitalizations, improves recovery, and saves costs. Its effectiveness is measured through doctors’ satisfaction, and overall healthcare efficiency.

### 4.10. Comparison with Published ML Articles

Machine learning (ML) models have been extensively applied to predict TBI prognosis, with mortality prediction remaining the most common trend. Wu and Lai (2023), as well as Courville (2023), demonstrated in their meta-analyses that machine learning significantly outperforms traditional methods in predictive accuracy [55,56]. Beyond mortality prediction, Fang C et al. (2022) employed ML to predict hospital length of stay [57]. Appiah Balaji NN et al. (2023) used ML to assist in predicting outcomes for TBI patients in rehabilitation hospitals [58]. Say I et al. (2022) revealed that ML surpasses traditional methods in accuracy, offering better predictions of functional independence during recovery after rehabilitation [59]. Van Deynse H et al. (2023) extended ML applications to predict patients’ likelihood of returning to work one year post-injury [60]. Matsuo K et al. (2023) utilized ML to classify discharge outcomes into three categories: functional recovery, disability, or death [61]. Meanwhile, Satyadev N et al. (2022) proposed an ML model to predict discharge placement, limited to mild and moderate TBI cases. Their model categorized outcomes as good, poor, or mortality, but did not address transitions to long-term care facilities [62]. Therefore, our study is novel in providing clinically meaningful predictive capabilities, offering early insights into long-term care trends for TBI patients.

### 4.11. Strength and Limitation

Our study possesses several strengths. First, it adheres to the Transparent Reporting of a Multivariable Prediction Model for Individual Prognosis or Diagnosis (TRIPOD) standard, enhancing the transparency and reproducibility of the research. Second, the use of various machine learning models (logistic regression, Random Forest, LightGBM, XGBoost, and MLP neural networks demonstrates a comprehensive approach to prediction modeling, thereby increasing the robustness of the findings. Third, techniques such as cross-validation and the Synthetic Minority Oversampling Technique (SMOTE) were employed to address issues like overfitting and class imbalance, respectively, enhancing the reliability of the models. Fourth, the study utilized a rigorous process for feature selection, considering clinical significance, statistical significance, and expert judgment, which strengthens the relevance of the chosen variables. Fifth, the study used multiple performance metrics, including accuracy, sensitivity, specificity, and Area under the Curve (AUC), providing a comprehensive evaluation of the machine learning models. The use of SHAP analysis to explain the importance of each feature in the machine learning models enhances the interpretability of the results. Finally, the integration of the developed model into the hospital system for real-world clinical applications demonstrates a practical translation of the research findings.

However, several limitations should be acknowledged. First, the retrospective design of the study may introduce biases and limit the establishment of causal relationships. Second, the study relies solely on electronic medical records from a single medical center in Taiwan, potentially restricting the generalizability of the findings to other populations or healthcare settings. Third, despite the application of SMOTE, the dataset still displays an imbalance in the outcome class, with more negative cases (no transfer to a long-term care center) than positive cases (transfer to a long-term care center). This imbalance may affect the generalizability and performance of the machine learning models. Employing a larger dataset or external validation could help alleviate this concern. In the future, we plan to develop an AI cloud service platform to share our model with other hospitals, using federated learning to predict prognosis. Fourth, temporal factors, such as changes in healthcare practices and patient demographics over time, are not considered in this study, potentially impacting the adaptability of findings to current clinical settings. Fifth, we acknowledge the importance of including non-survivors in the analysis. However, we have established a mortality predictive model in ICU cases [54]. The primary focus of this study was on predicting the need for long-term care in survivors of TBI. The rationale for excluding non-survivors is based on our specific objective: to assist healthcare providers in making decisions about post-discharge care for patients who survive the acute phase. In future work, we plan to develop a comprehensive model that stratifies patients into three categories: likely to recover and return home, likely to require long-term care, and likely to succumb during the acute phase. This integrated approach will ensure the model is applicable across the full spectrum of TBI outcomes. Finally, we recognize the importance of distinguishing between institutional and home-based care. However, in our computer system, we have not established neurological functioning and a socioeconomic system. In future studies, we will explore predictors that differentiate between these care settings, incorporating metrics such as neurological functioning, caregiver availability, and socioeconomic factors.

Therefore, in the future, we plan to include a broader range of data over extended periods, to understand how shifts in healthcare practices and patient demographics may influence our models’ performance and adaptability. Fifth, we have not evaluated the length of hospital stay [30], the Functional Status Score for the ICU [31], and the Coma Recovery Scale—Revised (CRS-R) score at discharge, which were proven predictors of RH. In the future, we plan to conduct further dynamic and continuous analysis, incorporating relevant parameters.

## 5. Conclusions

In summary, the study highlights the clinical effectiveness of the Random Forest model with 11 features. This model provides valuable early insights into post-traumatic brain-injury care trends, allowing for proactive arrangements for institutional or respiratory care ward support. This finding underscores the practical application of the model in clinical decision-making.

## Figures and Tables

**Figure 1 diagnostics-15-00020-f001:**
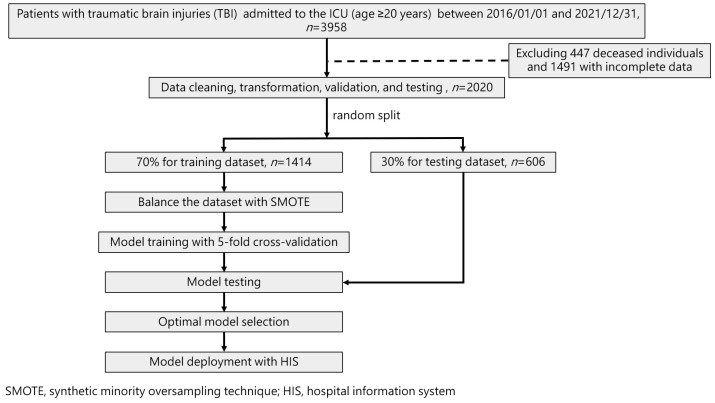
Flowchart showing the ML training process and its integration into the hospital system for TBI patients in ICU.

**Figure 2 diagnostics-15-00020-f002:**
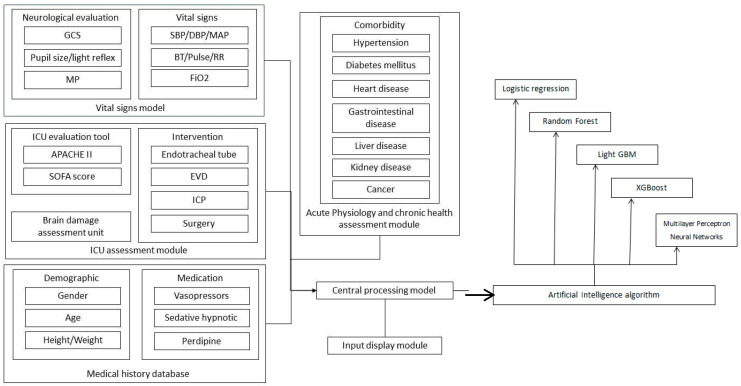
AI prediction device for TBI in ICU, insight into the system architecture and modules.

**Figure 3 diagnostics-15-00020-f003:**
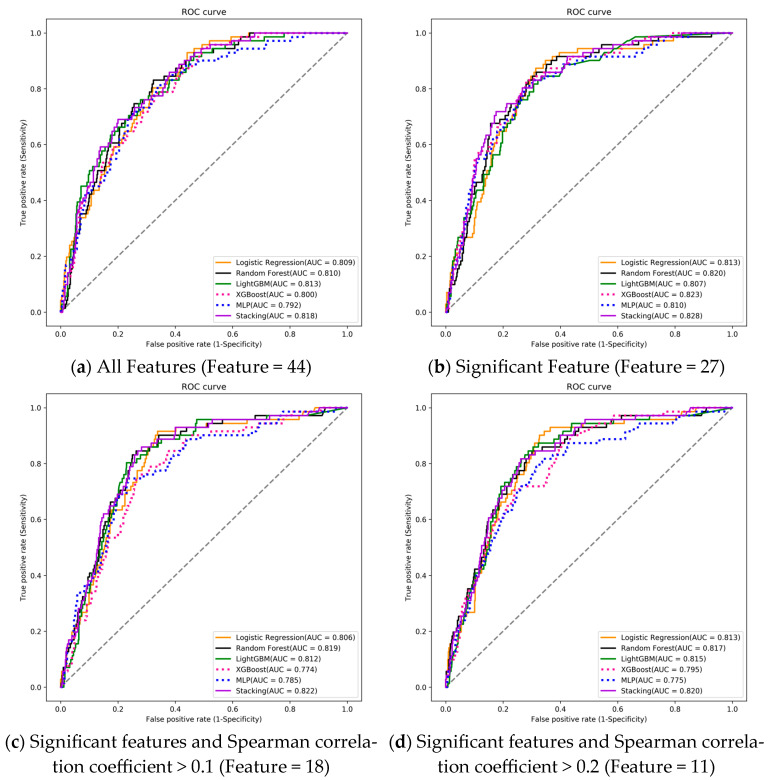
Receiver operating characteristic curves (ROC), area under the curve (AUC), for transfer to institute or respiratory care center prediction: (**a**) 44 features to train the ML model; (**b**) 27 features which were significant in transfer to institute or respiratory care center; (**c**) 18 features which were significant and Spearman correlation coefficient > 0.1; (**d**) 11 features which were significant and Spearman correlation coefficient > 0.2. Logistic Regression (orange), Random Forest (black), LightGBM (green), XGBoost (pink), and stacking (purple).

**Figure 4 diagnostics-15-00020-f004:**
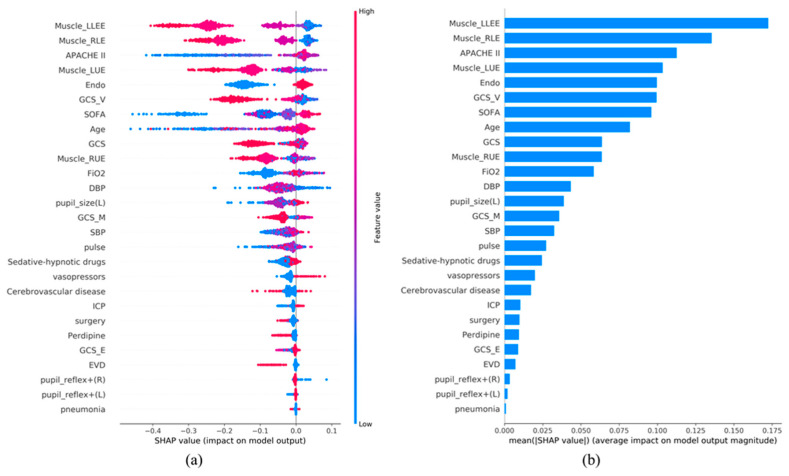
SHAP Analysis Results: (**a**) SHAP global explanation on the 27-feature model (XGBoost model); (**b**) SHAP absolute value of each feature on the 27-feature model (XGBoost model).

**Figure 5 diagnostics-15-00020-f005:**
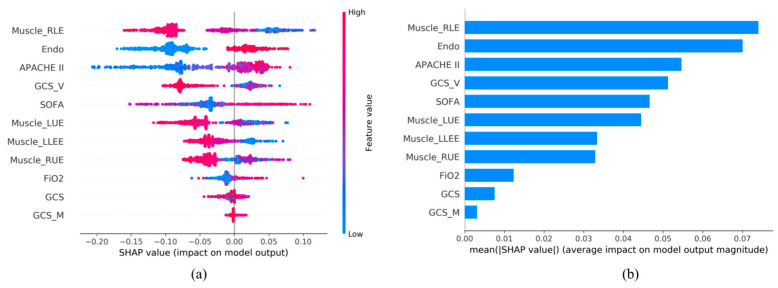
SHAP Analysis Results: (**a**) SHAP global explanation on the 11-feature model (Random Forest model); (**b**) SHAP absolute value of each feature on the 11-feature model (Random Forest model).

**Figure 6 diagnostics-15-00020-f006:**
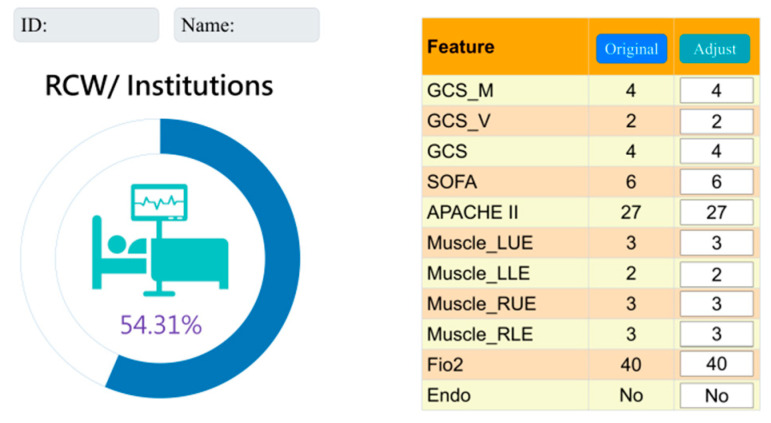
Interface presentation of AI in practical application within the Chi Mei Hospital healthcare system.

**Table 1 diagnostics-15-00020-t001:** (**1**) Characteristics and clinical presentations of patient transfer to long-term care center with traumatic brain injury. (**2**) Characteristics and clinical presentations of patient transfer to long-term care center with traumatic brain injury.

Feature	Overall	Long-Term Care Center		*p*-Value
		No	Yes	
	N = 2020	N = 1784	N = 236	
(**1**)				
Female, n (%)	735 (36.39)	663 (37.16)	72 (30.51)	0.054
Male, n (%)	1285 (63.61)	1121 (62.84)	164 (69.49)	
Age, mean (SD)	63.26 (17.76)	62.69 (18.08)	67.52 (14.52)	<0.001
Height, mean (SD)	162.75 (10.95)	162.67 (11.18)	163.37 (9.00)	0.277
Weight, mean (SD)	63.24 (14.23)	63.24 (14.10)	63.23 (15.27)	0.995
Systolic blood pressure (SBP), mean (SD)	143.33 (28.40)	142.56 (27.13)	149.15 (36.17)	0.007
Diastolic blood pressure (DBP), mean (SD)	78.72 (16.36)	79.08 (16.12)	76.02 (17.83)	0.013
Mean Arterial Pressure (MAP), mean (SD)	100.99 (19.86)	100.99 (19.40)	100.98 (23.06)	0.992
Body temperature (BT), mean (SD)	36.57 (0.56)	36.58 (0.55)	36.51 (0.62)	0.098
Pulse, mean (SD)	85.93 (15.90)	85.60 (15.64)	88.43 (17.53)	0.019
Respiratory rate (RR), mean (SD)	17.73 (3.95)	17.78 (3.93)	17.36 (4.06)	0.13
Glasgow Coma Scale_eye-opening (GCS_E), mean (SD)	3.31 (1.15)	3.40 (1.08)	2.61 (1.40)	<0.001
Glasgow Coma Scale_verbal response (GCS_V), mean (SD)	3.75 (1.66)	3.92 (1.57)	2.47 (1.72)	<0.001
Glasgow Coma Scale_motor response (GCS_M), mean (SD)	5.21 (1.60)	5.35 (1.49)	4.21 (2.00)	<0.001
Glasgow Coma Scale (GCS), mean (SD)	12.27 (4.11)	12.66 (3.86)	9.29 (4.70)	<0.001
Left Pupil				
Pupil reflex (−), n (%)	104 (5.15)	76 (4.26)	28 (11.86)	<0.001
Pupil reflex (+), n (%)	1916 (94.85)	1708 (95.74)	208 (88.14)	
Pupil size (L), mean (SD)	3.10 (0.77)	3.09 (0.73)	3.25 (1.01)	0.018
Right Pupil				
Pupil reflex (−), n (%)	103 (5.10)	74 (4.15)	29 (12.29)	<0.001
Pupil reflex (+), n (%)	1917 (94.90)	1710 (95.85)	207 (87.71)	
Pupil size (R), mean (SD)	3.09 (0.76)	3.08 (0.72)	3.16 (0.96)	0.26
Muscle power_left upper extremity (Muscle_LUE), mean (SD)	3.24 (1.54)	3.39 (1.46)	2.14 (1.70)	<0.001
Muscle power_left lower extremity (Muscle_LLE), mean (SD)	3.13 (1.58)	3.29 (1.50)	1.90 (1.65)	<0.001
Muscle power_right upper extremity (Muscle_RUE), mean (SD)	3.25 (1.54)	3.39 (1.46)	2.19 (1.73)	<0.001
Muscle power_right lower extremity (Muscle_RLE), mean (SD)	3.14 (1.58)	3.31 (1.49)	1.89 (1.64)	<0.001
(**2**)				
Inspired fraction of oxygen (FiO2), mean (SD)	26.49 (9.08)	25.86 (8.63)	31.25 (10.84)	<0.001
APACHE II, mean (SD)	11.71 (6.44)	11.11 (6.23)	16.31 (6.14)	<0.001
Sequential Organ Failure Assessment (SOFA score), mean (SD)	2.64 (2.26)	2.41 (2.10)	4.40 (2.61)	<0.001
Endotracheal tube (Endo)				
No, n (%)	1229 (60.84)	1158 (64.91)	71 (30.08)	<0.001
Yes, n (%)	791 (39.16)	626 (35.09)	165 (69.92)	
External ventricular drain (EVD)				
No, n (%)	1823 (90.25)	1627 (91.20)	196 (83.05)	<0.001
Yes, n (%)	197 (9.75)	157 (8.80)	40 (16.95)	
EVD_days, mean (SD)	5.01 (2.65)	4.85 (2.40)	5.62 (3.41)	0.184
Intracranial pressure (ICP), n (%)				
No, n (%)	1835 (90.84)	1652 (92.60)	183 (77.54)	<0.001
Yes, n (%)	185 (9.16)	132 (7.40)	53 (22.46)	
Surgery, n (%)	247 (12.23)	185 (10.37)	62 (26.27)	<0.001
Drugs				
Vasopressors, n (%)	157 (7.77)	106 (5.94)	51 (21.61)	<0.001
Sedative_hypnotic, n (%)	787 (38.96)	647 (36.27)	140 (59.32)	<0.001
Perdipine, n (%)	295 (14.60)	240 (13.45)	55 (23.31)	<0.001
Medical history				
Hypertension, n (%)	829 (41.04)	723 (40.53)	106 (44.92)	0.223
Diabetes mellitus, n (%)	510 (25.25)	443 (24.83)	67 (28.39)	0.27
Heart disease, n (%)	320 (15.84)	283 (15.86)	37 (15.68)	1
Cerebrovascular disease, n (%)	181 (8.96)	149 (8.35)	32 (13.56)	0.012
Gastrointestinal disease, n (%)	151 (7.48)	137 (7.68)	14 (5.93)	0.408
Liver disease, n (%)	135 (6.68)	122 (6.84)	13 (5.51)	0.529
kidney disease, n (%)	100 (4.95)	86 (4.82)	14 (5.93)	0.562
Cancer, n (%)	97 (4.80)	84 (4.71)	13 (5.51)	0.705
Thyroid disease, n (%)	53 (2.62)	46 (2.58)	7 (2.97)	0.894
Epilepsy, n (%)	40 (1.98)	35 (1.96)	5 (2.12)	0.804
Asthma, n (%)	39 (1.93)	31 (1.74)	8 (3.39)	0.123
Pneumonia, n (%)	32 (1.58)	24 (1.35)	8 (3.39)	0.027

**Table 2 diagnostics-15-00020-t002:** The Spearman correlation coefficient (r) for 44 features.

Feature	r	Feature	r	Feature	r	Feature	r
**APACHE II**	**0.259**	*Vasopressors*	*0.188*	Perdipine	0.09	DBP	−0.063
**SOFA**	**0.259**	*ICP*	*0.168*	EVD	0.088	BT	−0.053
**FiO2**	**0.232**	*Surgery*	*0.156*	Age	0.08	RR	−0.046
**Endo**	**0.229**	*Sedative–hypnotic drugs*	*0.152*	Cerebrovascular disease	0.059	Liver Disease	−0.017
**Muscle_RLE**	**−0.279**	*GCS_E*	*−0.199*	Pneumonia	0.053	Weight	−0.013
**Muscle_LLE**	**−0.272**	*pupil_reflex +(R)*	*−0.119*	SBP	0.052	MAP	−0.011
**GCS_M**	**−0.268**	*pupil_reflex +(L)*	*−0.111*	Pulse	0.044	Heart disease	−0.002
**GCS_V**	**−0.266**			Gender	0.044	Liver disease	−0.017
**GCS**	**−0.255**			Asthma	0.039	Gastrointestinal disease	−0.021
**Muscle_LUE**	**−0.249**			pupil_size (L)	0.034	RR	−0.046
**Muscle_RUE**	**−0.235**			Hypertension	0.029	Liver Disease	−0.017
				Diabetes mellitus	0.026		
				Height	0.019		
				Kidney disease	0.016		
				Cancer	0.012		
				Thyroid disease	0.008		
				Pupil_size (R)	0.006		
				Epilepsy	0.004		

MAP: mean arterial pressure; RR: respiratory rate; SBP: systolic blood pressure; EVD: external ventricular drainage; ICP: intracranial pressure. *Italic text*: Absolute value greater than 0.1; **Bold text**: Absolute value greater than 0.2.

**Table 3 diagnostics-15-00020-t003:** Details of different combinations of features according to clinical significance and Spearman correlation coefficient.

Model Definition	Number of Features Involved	Features
All study features	44	Gender, Age, Height, Weight, SBP, DBP, MAP, BT, Pulse, RR, GCS_E, GCS_V, GCS_M, GCS, Left Pupil reflex, Left Pupil size, Right Pupil reflex, Right Pupil size, Muscle_LUE, Muscle_LLEE, Muscle_RUE, Muscle_RLE, APACHE II, SOFA score, FiO2, Endo, EVD, ICP, Surgery, Vasopressors, Sedative–Hypnotic, Perdipine, Hypertension, Diabetes Mellitus, Heart disease, Cerebrovascular disease, Gastrointestinal disease, Liver disease, Kidney disease, Cancer, Thyroid disease, Epilepsy, Asthma, Pneumonia
Clinical significance	27	Age, SBP, DBP, Pulse, GCS_E, GCS_V, GCS_M, GCS, Left Pupil reflex, Left Pupil size, Right Pupil reflex, Muscle_LUE, Muscle_LLEE, Muscle_RUE, Muscle_RLE, APACHE II, SOFA score, FiO2, Endo, EVD, ICP, Surgery, Vasopressors, Perdipine, Sedative–Hypnotic, Pneumonia, Cerebrovascular disease
*p* < 0.05		
Clinical significance *p* < 0.05 plus Spearman correlation coefficient > 0.1 or <−0.1	18	GCS_E, GCS_V, GCS_M, GCS, Left Pupil reflex, Right Pupil reflex, Muscle_LUE, Muscle_LLEE, Muscle_RUE, Muscle_RLE, APACHE II, SOFA score, FiO2, Endo, ICP, Surgery, Vasopressors, Sedative–Hypnotic
Clinical significance plus Spearman correlation coefficient > 0.2 or <−0.2	11	GCS_V, GCS_M, GCS, Muscle_LUE, Muscle_LLEE, Muscle_RUE, Muscle_RLE, APACHE II, SOFA score, FiO2, Endo

**Table 4 diagnostics-15-00020-t004:** Model performances with different features combinations. Highest AUC in each category is highlighted in bold.

(A)				
Algorithm (44)	Accuracy	Sensitivity	Specificity	AUC
Logistic Regression	0.721	0.718	0.721	0.809
Random Forest	0.748	0.732	0.750	0.810
**LightGBM**	**0.728**	**0.732**	**0.727**	**0.813**
XGBoost	0.713	0.718	0.712	0.800
MLP	0.723	0.718	0.723	0.792
Stacking	0.741	0.732	0.742	0.818
**(B)**				
**Algorithm (27)**	**Accuracy**	**Sensitivity**	**Specificity**	**AUC**
Logistic Regression	0.749	0.732	0.751	0.813
Random Forest	0.746	0.746	0.748	0.820
LightGBM	0.754	0.732	0.757	0.807
**XGBoost**	**0.752**	**0.761**	**0.751**	**0.823**
MLP	0.751	0.746	0.751	0.810
Stacking	0.751	0.746	0.751	0.828
**(C)**				
**Algorithm (18)**	**Accuracy**	**Sensitivity**	**Specificity**	**AUC**
Logistic Regression	0.734	0.732	0.735	0.806
**Random Forest**	**0.749**	**0.831**	**0.738**	**0.819**
LightGBM	0.772	0.775	0.772	0.812
XGBoost	0.736	0.732	0.736	0.774
MLP	0.762	0.746	0.764	0.785
Stacking	0.761	0.761	0.761	0.822
**(D)**				
**Algorithm (11)**	**Accuracy**	**Sensitivity**	**Specificity**	**AUC**
Logistic Regression	0.749	0.746	0.75	0.813
**Random Forest**	**0.752**	**0.746**	**0.753**	**0.817**
LightGBM	0.756	0.789	0.751	0.815
XGBoost	0.723	0.718	0.723	0.795
MLP	0.708	0.746	0.703	0.775
Stacking	0.764	0.761	0.764	0.820

**Table 5 diagnostics-15-00020-t005:** DeLong test of ML models with different feature combinations.

Algorithm	Accuracy	Sensitivity	Specificity	AUC	Delong Test
Feature = 11 (Random Forest)	0.752	0.746	0.753	0.817	-
Feature = 44 (LightGBM)	0.728	0.732	0.727	0.813	0.916
Feature = 27 (XGBoost)	0.752	0.761	0.751	0.823	0.618
Feature = 18 (Random Forest)	0.749	0.831	0.738	0.819	0.728

## Data Availability

Based on the privacy of patients within the Chi Mei Medical Center’s Health Information Network, the primary data underlying this article cannot be shared publicly. However, de-identified data will be shared on reasonable request to the corresponding author.

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
