# Peer review of "Predictive Modeling of Long-Term Care Needs in Traumatic Brain Injury Patients Using Machine Learning"

_diagnostics, 2024, doi:10.3390/diagnostics15010020_

Round 1

Reviewer 1 Report

Comments and Suggestions for Authors

I suggest introducing a paragraph in the Methods section to discuss feature groups that were used.

The first paragraph of section 2.3 explains that various ML models were used; however, it does not mention Random Forest, the one that achieved the highest performance.

The background in the abstract deserves a review to clarify the research objective. The same observation applies to conclusions.

Reviewer 2 Report

Comments and Suggestions for Authors

The proposed approach is a simple deployment of some not very recent ML approaches.

The related works section lacks many recent works and a discussion on the limitations.

There is no justification on the selected ML models.

The study on appropriate features through some feature ranking and selection approaches is required.

Include a table on model setting and hyper-parameters.

Include some more alternative such and ensemble of the models in the evaluations.

Perform stability and scalability analysis.

Also include a discussion and comparisons on overfitting and generalization issues of the models.

Perform run time analysis and comparison.

Reviewer 3 Report

Comments and Suggestions for Authors

The authors developed a machine learning model that is able to predict long-term care needs in traumatic brain injury patients. A version of the model was integrated with the hospital system.

The authors provided detailed description of the modeling process, and good effort for showing the integration with the hospital system. However, a number of major concerns remain.

1. The largest concern I have is with the exclusion of almost half (1938/3958) of the individuals from the initial dataset. Even though the main aim is to look at need for long term care, the number of people who did not survive the acute phase should not be discounted. In a simplistic view, those with TBI could be categorized into those who are less injured and could return home, those who survive but need long term care, and the most severe cases would not survive the hospital stay. In the current model it only considers the first two cases, but not the third. However, at the deployment (assuming ICU admission), the highest risk should not be disregarded. How would the AI model's prediction be relevant if it does not include the ability to predict survival?

2. Related to the point 1, the exclusion of data also includes those with missing values and extreme outlier. What are the parameters used to determine these exclusions? With many imputation techniques, the authors should attempt to keep the data with missing values. Outliers should also not be disregarded without clear rationale.

3. There have been many attempts at creating models to predict TBI outcomes. These include both short term and long term outcomes. The literature review presented was not comprehensive. From a simple search, the top references were all not included in the current study. Please do a more thorough literature review to include more relevant and recent references. Example review:

Machine learning algorithms for predicting outcomes of traumatic brain injury: A systematic review and meta-analysis - PMC

4. The specific need for looking at long term care as a differentiating factor from mortality is also not sufficiently compelling. The authors mentioned that it is defined as those who are "admitted to long term care center excluding death". Why is it important to predict this group in particular?  Most families are interested in the ability of the patient to fully recover, or how likely the patient is likely to live past 6 months, 1 year, or 5 years. In what situation is there a need for predicting admission to long term care center?

5. On a related note, how about those who need long-term care, but can be managed at home with a caregiver? Perhaps indicators such as neurological functioning at discharge would be a better outcome measure in terms of predicting the need for long term care arrangements, whether it is at a center or at home. 

6. Dimension reduction techniques (e.g. PCA) should be explored in cases with many (44) features to see if these techniques improves model performance. 

7. Section 3.7, though highly relevant, provided too few details. This section should be expanded and tied back to the motivation for creating such an AI model. What could this AI model be used for? How can it benefit clinicians and patients? The authors mentioned enhanced resource allocation and improved healthcare outcomes. Are these impact being measured in a structured and comprehensive manner? 

Round 2

Reviewer 2 Report

Comments and Suggestions for Authors

All comments are addressed.